# Chemically specific termination control of oxide interfaces via layer-by-layer mean inner potential engineering

H.Y. Sun[1], Z.W. Mao[1], T.W. Zhang[1], L. Han[1], T.T. Zhang[1], X.B. Cai [2], X. Guo [3], Y.F. Li[1], Y.P. Zang[1], W. Guo[1], J.H. Song[1], D.X. Ji[1], C.Y. Gu[1], C. Tang[1], Z.B. Gu[1], N. Wang [2], Y. Zhu[3], D.G. Schlom[4,5], Y.F. Nie [1] & X.Q. Pan[1,6]

Creating oxide interfaces with precise chemical specificity at the atomic layer level is desired for the engineering of quantum phases and electronic applications, but highly challenging, owing partially to the lack of in situ tools to monitor the chemical composition and completeness of the surface layer during growth. Here we report the in situ observation of atomic layer-by-layer inner potential variations by analysing the Kikuchi lines during epitaxial growth of strontium titanate, providing a powerful real-time technique to monitor and control the chemical composition during growth. A model combining the effects of mean inner potential and step edge density (roughness) reveals the underlying mechanism of the complex and previously not well-understood reflection high-energy electron diffraction oscillations observed in the shuttered growth of oxide films. General rules are proposed to guide the synthesis of atomically and chemically sharp oxide interfaces, opening up vast opportunities for the exploration of intriguing quantum phenomena at oxide interfaces.

[1] National Laboratory of Solid State Microstructures, Jiangsu Key Laboratory of Artificial Functional Materials, College of Engineering and Applied Sciences, and Collaborative Innovation Center of Advanced Microstructures, Nanjing University, 210093 Nanjing, China. [2] Department of Physics, Hong Kong University of Science and Technology, Clear Water Bay, Hong Kong, China. [3] Department of Applied Physics, The Hong Kong Polytechnic University, Hung Hom, Kowloon, Hong Kong, China. [4] Department of Materials Science and Engineering, Cornell University, Ithaca, NY 14853, USA. [5] Kavli Institute at Cornell for Nanoscale Science, Ithaca, NY 14853, USA. [6] Department of Chemical Engineering and Materials Science and Department of Physics and Astronomy, University of California, Irvine, 916 Engineering Tower, Irvine, CA 92697, USA. These authors contributed equally: H. Y. Sun, Z. W. Mao. Correspondence and requests for materials should be addressed to Y.F.N. (email: ynie@nju.edu.cn)

Oxide interfaces exhibit a rich variety of emergent phenomena, including two-dimensional electron liquids (2DEL), superconductivity, ferromagnetism, and anti-ferromagnetism[1–7]. These open promising vistas for oxide electronic devices, but the sample preparation demands unprecedentedly precise control of the atomic configuration of interfaces. For example, the lanthanum aluminate and strontium titanate (LaAlO$_3$/SrTiO$_3$) system only shows 2DEL at the n-type LaO/TiO$_2$ interface but not at the p-type AlO$_2$/SrO interface[1,2]. Moreover, strongly correlated electronic properties of oxides are typically extremely sensitive to the exact atomic configuration due to slight energetic differences between competing ground states[8,9]. Modern growth technique advances, especially the application of in situ reflection high-energy electron diffraction (RHEED), greatly improve the growth precision as one period of the RHEED intensity oscillations typically represents the growth of a charge-neutral formula unit[10–18]. By simply counting the number of periods in RHEED oscillations, one can control the film thickness with monolayer precision. Lacking, however, are reliable real-time in situ methods to monitor the chemical composition and to identify the completeness of the growth front layer during layer-by-layer growth of oxide films, hindering the synthesis of chemically precise oxide interfaces for compelling quantum phases. Considering the prototype perovskite SrTiO$_3$ as an example, the maxima(minima) of the RHEED intensity is typically assumed to correspond to the fully SrO- (TiO$_2$-) terminated surface[19–22], however, the opposite or more complicated situations are also commonly observed[19,23–25]. In fact, the origin of RHEED oscillations and why the period of the RHEED oscillations during the shuttered growth of perovskite (like SrTiO$_3$) films corresponds to the growth of two monolayers (one SrO monolayer and one TiO$_2$ monolayer)[19–22] instead of one monolayer like that in growing conventional semiconductors[26–28] is still not understood. This cannot be simply explained by the step edge density model (Fig. 1b), since both SrO and TiO$_2$ terminations are smooth. All of these give rise to a fundamental question if one can rely on RHEED intensity oscillations to precisely control the growth of oxide interfaces.

Since RHEED oscillations during the layer-by-layer growth of complex oxide films are complicated and sometimes controversial as reported in the literature[19,22–25], commonly accepted criteria for real-time control of the surface termination by RHEED is still lacking and it is very challenging to synthesize chemically sharp oxide interface. Although other in situ measurements, such as X-ray photoemission spectroscopy (XPS), can provide chemical information of the surface termination[20,29], most of these techniques need ultrahigh vacuum and not applicable in monitoring the chemical composition in real time during the growth. To date, 2DELs in LAO/STO have mainly been fabricated on chemically etched TiO$_2$-terminated SrTiO$_3$, post-annealed SrTiO$_3$ films, or silicon substrates[1,2,30,31]. Attempts of synthesising 2DEL on epitaxial SrTiO$_3$ films were shown to be extremely challenging since the SrTiO$_3$ films thicker than 7 unit cells (u.c.) exhibit imperfect (mixed) termination[32,33]. Ex situ etching and annealing after the growth, growing SrTiO$_3$ films in very high temperature (1100 °C), and using metal-organic precursors in hybrid molecular-beam epitaxy have been shown to achieve TiO$_2$-terminated surface[34–37], but these methods have limitations (only work for few oxides of certain terminations, non-water-sensitive oxides, etc.) and cannot be universally applied to other oxide systems. Therefore, understanding the origin of RHEED oscillations and finding in situ tools for characterising the chemical composition and completeness of the surface layer are highly desired in order to synthesise atomically sharp oxide interface for intriguing quantum phases and electronic applications.

In this work, we report the in situ measurements of the mean inner potential using Kikuchi lines and the observation of atomic layer-by-layer variations of the mean inner potential during the growth of SrTiO$_3$, which provide information on the chemical composition of the film surface layer. The underlying mechanism

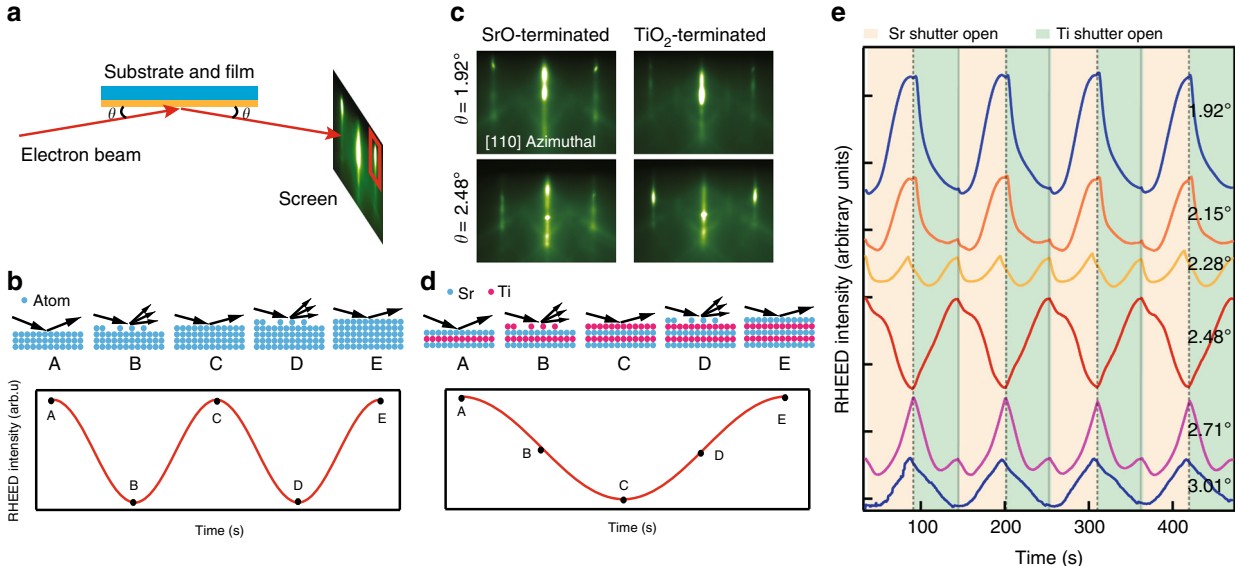

**Fig. 1** Phase inversion and frequency doubling of RHEED oscillations during the growth of SrTiO$_3$ films. **a** Schematic of a typical RHEED system. **b** In a simple model, the diffuse scattering from periodic surface roughing results in the variation of the RHEED intensity. Its period corresponds to the growth of a monolayer of material. **c** RHEED patterns of SrTiO$_3$ films taken along the [110] azimuthal direction typically show maximum (minimum) diffraction intensity in the case of SrO- (TiO$_2$-) terminated surfaces, but this surface termination dependence can be reversed by changing the incident angle of the electron beam. **d** RHEED intensity oscillations of (11) diffraction peaks during the atomic layer-by-layer growth of SrTiO$_3$ films shows a period corresponding to two monolayers, which cannot be explained by the simple step edge density model. **e** The intensity of the (11) diffraction peak oscillates during the atomic layer-by-layer growth of SrTiO$_3$ films and its period and phase strongly depend on the incident angle of the electron beam

for the complex RHEED intensity oscillations in growing oxide films is also uncovered by including the effects of mean inner potential and the inelastic scattering by the step edges. General rules for synthesising atomically sharp oxide interfaces are also proposed and demonstrated by realising 2DEL on thick homo-epitaxial SrTiO₃ films.

## Results

**Epitaxial growth and characterization.** Epitaxial SrTiO₃ films were grown on (001) SrTiO₃ single crystalline substrates by reactive molecular-beam epitaxy (MBE) equipped with an in situ RHEED system. SrTiO₃ substrates were chemically etched to yield pure TiO₂-terminated surface using the standard method reported previously[38]. A shutter-controlled technique was used to deposit alternating monolayers of SrO and TiO₂. The quality of film surfaces and interfaces was also characterized by other techniques, including high-resolution transmission electron microscopy (HRTEM), X-ray diffraction (XRD), XPS, atomic force microscopy (AFM), and lateral force microscopy (LFM). All of the data are of high quality and consistent with our conclusions. More details about film growth, structural characterizations, RHEED data analysis, and simulations are provided in the Supplementary Notes 1, 4 and 5, Supplementary Figures 1–7.

**Observation of phase inversion and frequency doubling.** RHEED oscillations in growing oxide films are very complicated and previously reported observations are controversial[19–21,23–25,39]. In this article, we found that all these complicated types of RHEED oscillations can be reproduced by just changing the measurement parameters (i.e. the incident angle of the electron beam) without changing any growth parameters. In Fig. 1, we show the observation of an intriguing phase inversion and frequency doubling of the RHEED oscillations in the atomic layer-by-layer growth of SrTiO₃ films by only varying the incident angle of the electron beam. Initially, we optimise the growth to obtain the typical RHEED oscillation pattern of the (11) diffraction peaks, showing a maximum (minimum) intensity at the SrO- (TiO₂-) terminated surface, which is consistent with many previous observations[19–22]. As the electron beam incident angle increases, the amplitude of the RHEED oscillations decreases and a small peak develops at the minima. A clear frequency doubling feature with smallest oscillation amplitude is observed at a higher incident angle (~2.28°). Further increasing the incident angle, the RHEED oscillation amplitude increases again and gradually develops into a phase inverted (180° phase difference) oscillation pattern. This complicated incident angle dependence of the RHEED oscillations reconciles the contradicting observations reported in the literature[19–21,23–25]. It is important to note that the minimum RHEED intensity does not correspond to a complete termination in most cases, in sharp contrast to the widely used guidelines of terminating the film growth at the minima to obtain fully TiO₂-terminated surface.

Similar phase inversion and frequency doubling are also observed in LaFeO₃ and other oxide systems (see Supplementary Note 3 and Supplementary Fig. 4). This indicates that the phase inversion and frequency doubling in oxide film growth are not limited to specific types of elements or polarity of the crystalline structure, but from a common origin.

**The mean inner potential variations during oxide growth.** In addition to the RHEED intensity oscillations, we also observe intriguing variations of the position of the Kikuchi lines (curved

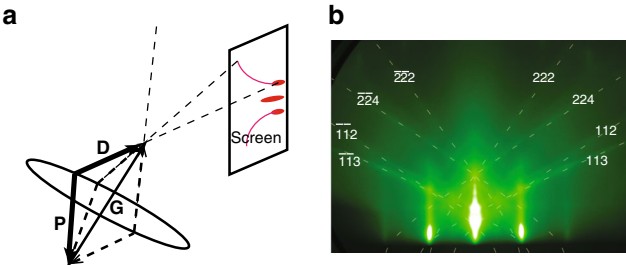

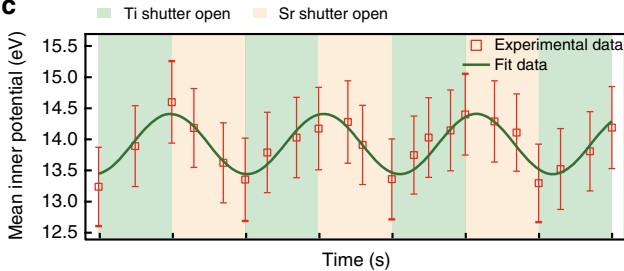

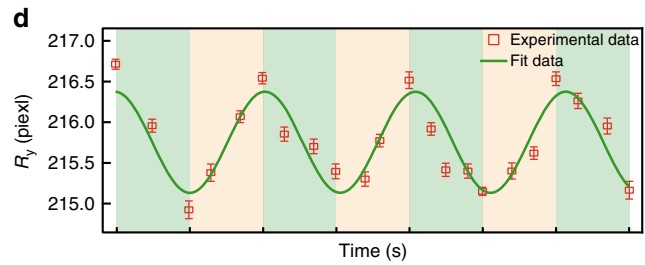

**Fig. 2** Kikuchi lines and atomic layer-by-layer variations of the mean inner potential. **a** Schematic of the formation of Kikuchi lines. **G** is the reciprocal vector of the lattice, **P** and **D** are the momentum vectors of the primary incident electron beam and the outgoing beam that satisfy the Bragg condition. The red dots and lines on the screen are the diffraction peaks and the Kikuchi lines, respectively. **b** A typical RHEED pattern taken along the SrTiO₃ [110] direction, showing Kikuchi lines clearly. The dashed lines are fits to the Kikuchi lines using Eq. (1). **c** Periodic variation of the extracted mean inner potential. The error bars are the fitting error of the Kikuchi lines corresponding to **G**(224) and **G**(113) in each RHEED pattern using Eq. (15) in the Supplementary Note 6. **d** Position shift $R_y$ of the Kikuchi lines and the small error bars indicate clear and reliable relative change of the mean inner potential during the growth (Supplementary Fig. 7 and Supplementary Note 7). The error bars are the fitting error of the Kikuchi lines using Guassian function

lines in the RHEED spectra) during the growth of SrTiO₃ films. Incident electrons that have undergone diffuse scattering prior to undergoing a Bragg reflection give rise to the curved lines (Kikuchi bands) on the screen (Fig. 2a). Kikuchi lines are independent of the direction of the original incident beam but depend only upon the crystal orientation, the electron momentum and the mean inner potential[40]. The mean inner potential is a property of materials related to the surface dipole and defined as the volume average of the atomic electrostatic potentials in the specimen. As the surface dipole strongly depends on the surface composition, the mean inner potential can provide information about the composition of the surface termination.

When the electrons pass the surface and enter the film, the electrons feel attraction due to the mean inner potential, resulting in an increase of the electron momentum that is perpendicular to the surface (Fig. 3a). Therefore, if the mean inner potential varies

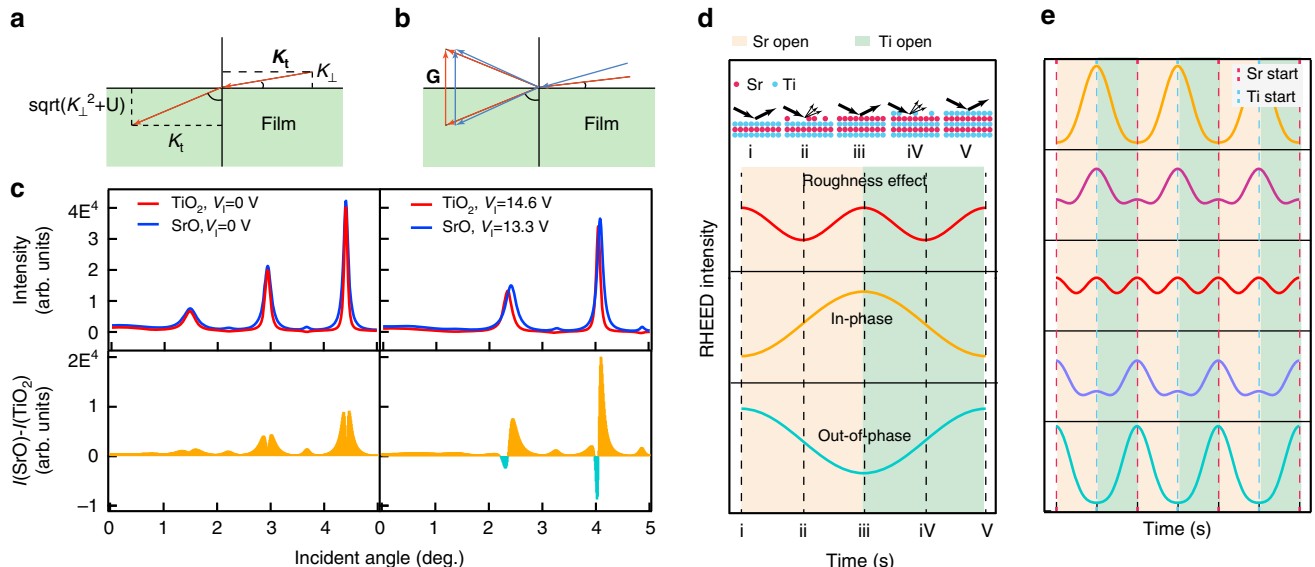

**Fig. 3** Origin of RHEED oscillations. **a** The mean inner potential results in an increase of the component of the electron momentum that is perpendicular to the film surface. **b** Higher mean inner potential (blue lines) will decrease the incident angle that satisfy the Bragg diffraction condition. **c** The calculated electron diffraction intensity as a function of incident angle shows a strong dependance on the mean inner potential. For simplicity, when SrO (TiO₂) termination yields stronger diffraction intensity, it is called the in-phase (out-of-phase) condition. **d** Schematic of three extreme cases of RHEED oscillations that include only periodic surface roughing, pure in-phase, and pure out-of-phase oscillations. **e** Simulated RHEED oscillations calculated by combining the three extreme case components shown in panel (**d**) with different weighting factors to simulate the incident angle dependence of the electron beam

during the film growth, the diffraction condition and position of the Kikuchi lines will also vary. The Kikuchi line for the set of planes corresponding to the reciprocal lattice vector $\mathbf{G}(hkl)$ is given by[40]

$$\mathbf{K} \cdot \mathbf{G} = K_t \cdot G_t + \left(K_\perp^2 + U\right)^{1/2} G_\perp = \frac{|\mathbf{G}|^2}{2},$$
$$\left(U = \left(2m_e e/\hbar^2\right) V_I\right)$$

(1)

where $\mathbf{K}$ is the momentum of electrons, $K_t$ ($K_\perp$) is the component of the momentum parallel (normal) to the surface, $V_I$ is the mean inner potential and $\mathbf{G}$ is the reciprocal lattice vector.

By fitting the position of the Kikuchi lines of $\mathbf{G}(224)$ and $\mathbf{G}(113)$ using Eq. (1), the mean inner potential $V_I$ can be extracted (Fig. 2b) (see Supplementary Note 6 for details). Remarkably, the mean inner potential oscillates during the atomic layer-by-layer growth of SrTiO₃ films (Fig. 2c), indicating that the mean inner potential of SrO and TiO₂ terminations are different. The extracted mean inner potential of SrO- (TiO₂-) terminated surface is 13.3 V (14.6 V), which are within the range of reported values from photoemission spectroscopy measurements[41–43]. Note that the error bar of the absolute mean inner potential (Fig. 2c) is relatively large. This large systematic error is because the lineshape of the Kikuchi lines could not be perfectly accounted by the ideal equation due to the imperfect diffraction geometry and distortion of the RHEED images taken by the CCD camera. Nonetheless, the position shift of the Kikuchi lines (Fig. 2d) also oscillate during the growth and the small fitting error bar indicates the relative change of the mean inner potential is clear and reliable (see Supplementary Note 7 for details). The difference of mean inner potentials between these two terminations is about 1.3 V, comparable with the reported work function difference from in situ Kelvin probe measurements[44] and recent theoretical calculations[45].

This direct in situ measurement of the mean inner potential during film growth can provide important information on the chemical composition of the surface layer during the growth. For example, it can be used to precisely control the surface and

interface of oxides. In principle, this technique should also work for other film growth techniques like pulsed-laser deposition (PLD) and sputtering when RHEED is measured during growth[46,47].

**Origin of RHEED oscillations.** To understand the origin of RHEED oscillations, we first discuss the mechanism of the observed phase inversion of RHEED oscillations within the framework of kinematic diffraction theory. Electron diffraction is fundamentally a dynamical effect involving multiple scattering processes[40]. The complex algebra involving the recursion of transfer scattering matrix makes the simulation using dynamical theory a rather difficult task and it is also difficult to visualize the key underlying physics, such as the role of mean inner potential. Considering we can directly measure the mean inner potential, kinematic theory is the more suitable model to investigate the role of mean inner potential in determining the RHEED intensity. In the following discussion, we concentrate on the shift of the incident angle that satisfies the diffraction condition due to the mean inner potential variation and the resulting relative change of the diffraction intensity where the kinematic diffraction theory is valid. As such, we keep our discussion in a qualitative level within the framework of kinematic diffraction theory for simplicity.

To calculate the total diffracted beam intensity, the scattering amplitudes from all atoms within the probe depth are attenuated exponentially according to depth and summed and squared, yielding,

$$I = \left| \sum_j f_j \cdot \exp\left(-i\Delta \mathbf{K} \cdot \mathbf{r}_j\right) \cdot \exp\left(-z_j/(L \sin \theta)\right) \right|^2$$

(2)

where $f_j$ is the atomic scattering factor[40], for Sr it is 38, for Ti it is 22, and for O it is 8, $\Delta \mathbf{K}$ is the scattering vector of the beam inside the crystal, $\mathbf{r}_j$ is the position vector of the $j$ atom, $L$ is the electronic mean-free path, $z_j$ is the depth of the $j$ atom from the

surface, and $\theta$ is the incident angle. Note that $\Delta\mathbf{K}$ includes the contribution of the mean inner potential ($V_I$). More details can be found in the Supplementary Notes 4–6. In our experiments, the electron energy is 15 keV and its mean free path is about 11 nm according to the NIST database. The electron inelastic-mean-free-paths were estimatedusing NIST Standard Reference Database71, NIST Electron Inelastic-Mean-Free-PathDatabase: Ver. 1.2. It is distributed via the website https://www.nist.gov/srd/nist-standard-referencedatabase-71, and references therein.

In Fig. 3c, we plot the simulated diffraction intensity as a function of the electron beam incident angle for both SrO-terminated and TiO$_2$-terminated surfaces with and without considering the mean inner potential. Overall, the diffraction intensity from the SrO-terminated film is slightly higher because the electron density of the SrO sublayer is higher than that of the TiO$_2$ sublayer. Note that the electron momentum $\mathbf{K}$ varies when electrons enter or leave the film surface, since they feel the attraction or repulsion by the surface dipole (described by the mean inner potential $V_I$), resulting in an increase of the component of electron momentum perpendicular to the surface (Fig. 3a). Taking the mean inner potentials ($V_I$) into account, the normal component of electron momentum increases, resulting in a decrease of the incident angle in order to satisfy the diffraction condition (Fig. 3b).

Using the mean inner potentials extracted from the fits to the position of the Kikuchi lines, a clear offset of the diffraction curve is shown between SrO-terminated and TiO$_2$-terminated SrTiO$_3$, resulting in multiple sign switchings in the diffraction intensity difference plot (Fig. 3c). During the atomic layer-by-layer growth of SrTiO$_3$, the surface termination switches between SrO and TiO$_2$ sublayers, resulting in intensity oscillations with a period corresponding to the growth of two sublayers. In the regime shaded in orange (cyan) in the diffraction intensity difference plot, the RHEED intensity increases when depositing the SrO (TiO$_2$) layer since the SrO- (TiO$_2$-) terminated SrTiO$_3$ has higher diffraction intensity than the other termination.

When the incident angle of the electron beam crosses the sign switching point where both SrO and TiO$_2$ termination yield the same diffraction intensity, the relative intensity of these two terminations is switched and the phase of RHEED oscillations is thus inverted (180° phase shift). For the sake of simplicity in the following discussion, we define the oscillation pattern with increasing diffraction intensity during the deposition of SrO (TiO$_2$) as in-phase (out-of-phase) oscillations. The location of the calculated sign switching points matches our experimental data shown in Fig. 1e qualitatively. Moreover, the significant decrease of the RHEED oscillation amplitude when the electron beam incident angle approaches sign switching points is consistent with this model. Note that the observation of this 180° phase shift is distinct from the continuous phase shift observed in the growth of the conventional semiconductor Si (111), which is explained by a simple potential model[40].

We now turn to the mechanism of the frequency doubling feature. At the phase switching angle where the frequency doubling feature dominates the oscillation, the diffraction intensity reaches its maxima at the end of the deposition of each sublayer and reaches its minima in the middle of the deposition of each sublayer (Fig. 1e), implying that the frequency doubling is most likely caused by the inelastic scattering by the periodic surface roughing, similar to the step density model. In the step density model, the diffraction intensity decreases when growth roughens the surface and then increases again as the islands coalesce and reach a maximum when the surface layer is completed[10–12,14–18].

All together, there are two major contributions to the RHEED oscillations. The offset of the diffraction curves caused by the variation of the mean inner potential gives rise to the 180° phase inversion of the RHEED oscillations, while the periodic surface roughing is responsible for the frequency doubling feature. By varying the weights of these two contributions, we simulate the incident angle dependence of the RHEED oscillations (Fig. 3d, e), which qualitatively explains the observed RHEED intensity oscillations (Fig. 1e).

**General guidelines for precise surface and interface growth.** Our results and analysis show that the conventional guidelines of terminating the film growth at the minima to obtain a TiO$_2$-terminated SrTiO$_3$ only works under conditions where the roughness scattering contribution is minimised. Based on above model, we propose three general guidelines for the growth of complex oxides, such as perovskites ($ABO_3$), with chemically specific termination control.

Rule 1: A technique that deposits each sublayer separately in a sequence is preferred. In growing oxide films using a co-deposition technique where all elemental species are deposited simultaneously, the amplitude of the RHEED oscillations typically decreases gradually[25,34,48,49]. From our model, the oscillation amplitude mainly corresponds to the diffraction intensity difference between two distinct surface terminations and the decrease of the amplitude indicates a mixture of the terminations. The RHEED oscillations during the film growth using the shuttered method by MBE typically hold constant amplitude, indicating the preservation of sharp termination throughout the whole film growth process. By using multiple targets, other film growth technique like PLD has also recently been able to yield stable RHEED oscillations with constant amplitude throughout the film growth[23,50].

Rule 2: The larger the amplitude of the RHEED oscillations by optimizing the incident angle of the electron beam, the easier to obtain precise control of the termination. As shown in the experimental data and in the model simulation, the oscillation amplitude strongly relies on the incident angle of the electron beam. At certain incident angles, two different terminations yield similar diffraction intensity where the total oscillation amplitude is small and the frequency is doubled. Thus, the minima do not correspond to any full terminations. By optimizing the incident angle of the electron beam to maximize the total oscillation amplitude, both maxima and minima correspond to pure terminations, and it is easier to obtain more precise termination control.

Rule 3: More precise termination control can be achieved by terminating the growth at RHEED maxima. As shown in Figs. 1e and 3e, the minima of the RHEED oscillation with the double valley feature correspond to random termination, but the maxima corresponding to chemically specific complete surface termination. Therefore, a better choice is to adjust the angle of incidence to the in-phase or out-of-phase configuration to terminate the growth at the maximum RHEED intensity in order to reliably obtain the desired surface termination.

For heteroepitaxial film growth, the initial oscillation patterns are typically complicated due to the different diffraction structure factor and mean inner potential. Only when the film is thicker than the mean free path of the electron beam, stable RHEED oscillation pattern can be achieved and the above method can be applied.

**Realising LaAlO$_3$/SrTiO$_3$ 2DEL on thick SrTiO$_3$ films.** To test the validity of our model, we grew the LaAlO$_3$/SrTiO$_3$ 2DEL on

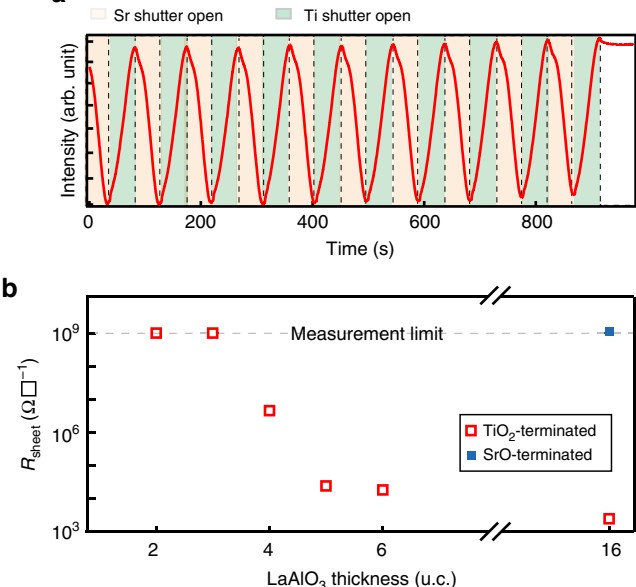

**Fig. 4** 2DEL of LAO/STO grown on epitaxial SrTiO₃ films. **a** Out-of-phase RHEED oscillations (only the last 10 cycles are shown) for the growth of TiO₂-terminated homoepitaxial SrTiO₃ films. **b** Room-temperature sheet resistivity of the n-type LAO/STO interfaces grown on 60 u.c. thick TiO₂-terminated epitaxial SrTiO₃ films, showing a critical LAO thickness of 4 u.c. The p-type LAO/STO interface grown on SrO-terminated epitaxial SrTiO₃ film is insulating (blue solid square)

thick epitaxial STO films. As mentioned previously, growing high quality TiO₂-terminated SrTiO₃ films for synthesising LaAlO₃/SrTiO₃ 2DEL has been shown to be extremely challenging[32,33]. This makes it an excellent testbed of the above general guidelines. Following above rules, we grew thick TiO₂-terminated epitaxial SrTiO₃ films on (001) SrTiO₃ substrates and deposited a LaAlO₃ layer of various thicknesses on top of it. The incident angle of the electron beam was tuned to obtain the out-of-phase RHEED oscillation pattern and the growth of SrTiO₃ films was terminated at the maximum RHEED intensity to obtain a complete TiO₂ surface termination (Fig. 4a). The LaAlO₃/SrTiO₃ films were grown in a distilled ozone atmosphere and annealed in 1 atm of pure oxygen after growth to avoid the formation of oxygen vacancies.

All SrTiO₃ films covered with 4 u.c. and thicker LaAlO₃ films are conducting while the SrTiO₃ films covered with thinner LaAlO₃ layer are insulating (Fig. 4b), indicating the formation of a 2DEL. We also prepared SrO-terminated homoepitaxial SrTiO₃ and the p-type LaAlO₃/SrTiO₃ interface turns out to be totally insulating as expected. More details can be seen in Supplementary Note 2 and Supplementary Figures 2 and 3. The capability of synthesising 2DEL on arbitrary thick SrTiO₃ epitaxial films underscores the validity and practical importance of our model and guidelines.

Note that high-quality oxide surface/interface and high mobility 2DEL can also be synthesized with special growth receipts including post annealing, high growth temperature, or metal-organic precursor, etc.[30–32,37]. Our guidelines provide a more convenient and general way to control the surface termination and oxide interface under normal layer-by-layer growth condition with the proper application of RHEED.

## Discussion

In conclusion, we report in situ measurements of the mean inner potential of complex oxides by fitting the Kikuchi lines

showing layer-by-layer variations of the mean inner potential during growth of SrTiO₃. This provides important chemical information about the surface termination during the growth in real time. We also observed a phase inversion and frequency doubling in the RHEED oscillations by only varying the incident angle of the electron beam, which reconcile the contradicting observations reported previously. A model including the effects of the mean inner potential and periodic surface roughing reveals the underlying mechanism of the complicated RHEED intensity oscillations in oxide growth. Based on this model, general guidelines are proposed for the growth of chemically precise oxide interfaces, paving the way to the exploration of emerging quantum phases at a wide range of oxide interfaces.

## Methods

**Single-terminated SrTiO₃ films growth**. SrTiO₃ substrates were etched in buffered 8% HF(NH₄F) acid for 45 s to achieve pure TiO₂ termination[38], followed by annealing at 950 °C in the furnace under flowing pure oxygen for 80 min to obtain sharp step-and-terrace surfaces. The single-terminated (including SrO-terminated or TiO₂-terminated) SrTiO₃ films grown on (001) SrTiO₃ substrates were prepared at 750 °C (measured by pyrometer) and $1 \times 10^{-6}$ Torr of distilled ozone using a DCA R450 MBE system equipped with an in situ RHEED system. An electron beam with 15 keV energy was used in the RHEED measurements. After a rough calibration of the source fluxes using a quartz crystal microbalance (QCM), the precise shutter times for Sr and Ti were calibrated using an efficient co-deposition method[25]. After the flux calibrations, a shutter-controlled method was used for the atomic layer-by-layer growth of SrTiO₃, where the shutters mounted in front of the Sr and Ti sources were alternately opened and closed during the growth. An out-of-phase growth mode was used to precisely obtain pure TiO₂-termination. For the typical in-phase (out-of-phase) growth, we first adjusted the incident angle of the electron beam to minimise (maximise) the intensity of (11) diffraction peaks of the TiO₂-terminated SrTiO₃ substrates.

**Epitaxial LaAlO₃ growth**. The LaAlO₃ layers were grown at 750 °C (measured by pyrometer) and $1 \times 10^{-6}$ Torr of distilled ozone, using the same calibration method as reported in the literature[51]. We prepared samples with different thickness of LaAlO₃ films (2–6, 8 and 16 u.c.) on 20 or 60 u.c. SrTiO₃ epitaxial films grown on SrTiO₃ substrates. As mentioned in the literature[52], nearly perfect stoichiometric LaAlO₃ films grown on SrTiO₃ show no 2DEL. Here, we just fixed the La/Al ratio to be 0.8 for all LaAlO₃ films to obtain 2DEL.

**Transport measurements**. The interfacial electronic properties of the LaAlO₃/SrTiO₃ heterostructure were probed using four-point electrical transport measurements (Quantum Design, Physical Properties Measurement System) using the Van der Pauw geometry.

**Scanning transmission electron microscopy (STEM) measurements**. STEM-high angle annular dark-field (HAADF) images were performed using JEOL JEM ARM 200F equipped with a cold field emission gun and an ASCOR fifth-order probe corrector. Simultaneous spectrum imaging of electron energy-loss spectroscopy (EELS) and X-ray energy-dispersive spectroscopy (EDS) was carried out under 200 kV accelerating voltage with a 26 mrad convergence angle for the optimal probe condition. Energy dispersion of 0.25 eV per channel and 91 mrad collection angle were set up for EELS and double large solid-state detectors from JEOL were used for EDS. The HAADF image was acquired with a 93 mrad inner angle simultaneously.

**Data availability**. The data that support the findings of this study are available from the authors on reasonable request, see author contributions for specific data sets.

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

## Acknowledgements

Authors are grateful to Chen Li and Di Wu (Nanjing University) for the help with AFM experiments and Hui Li (Nanjing University) for the assistance with XPS experiments. We gratefully acknowledge insightful discussions with Scott Chambers, Xiaoxing Xi, Gertjan Koster, Jiandong Guo, and Kyle Shen. This work was supported by the National Basic Research Program of China (Grant no. 2015CB654901), the National Natural Science Foundation of China (Nos. 11574135, 11774153, 51772143, 51672125), and the Fundamental Research Funds for the Central Universities (No. 0213-14380058). Y. F. N. is supported by the National Thousand-Young-Talents Program and Program for High-Level Entrepreneurial and Innovative Talents Introduction, Jiangsu Province. D. G. S. was supported by Army Research Office (ARO) grant W911NF-16-1-0315. X. B. C. and N. W. are supported by the Research Grants Council of Hong Kong (Project no. C6021-14E) and the WMINST Grant for the joint research between HKUST and NJU National Laboratory of Solid State Microstructures. X. Y. G., Y. Z. are supported by the Hong Kong Research Grants Council through the Early Career Scheme (Project no. 25301617) and the Hong Kong Polytechnic University grant (Project no. 1-ZE6G).

## Author contributions

Z. B. G., Y. F. N., and X. Q. P. conceived and designed the experiments. D. G. S. provided advice at multiple stages of this research. Z. W. M. grew the samples and took the RHEED data for mean inner potential simulation with the help of T. W. Z., W. G., D. X. J., C. Y. G., and C. T., and performed the transport experiments and analysed the data. H. Y. S. rechecked the mean inner potential simulation and theoretical calculations and grew single-terminated SrTiO₃ films for ex situ characterizations with the help of T. T. Z., Y. P. Z., and J. H. S. H. Y. S. prepared the RHEED oscillations of LaFeO₃ with the help of C. Y. G. and wrote the Supplementary Information. L. H. and Y. F. L. prepared the single-terminated substrates and performed the AFM measurements and data analysis. X. G., X. B. C., N. W., and Y. Z. contributed to the STEM measurements. Y. F. N., Z. W. M., and

H. Y. S. wrote the manuscript. All authors discussed the data and contributed to the manuscript.

## Additional information

**Competing interests:** The authors declare no competing interests.

