## [Peer Review File · Nature Communications]

Reviewers' Comments:

Reviewer #1:

Remarks to the Author:

The authors report on RHEED experiments done in different epitaxial oxide structures. The authors pay particular attention to the oscillation patterns obtained during the growth of the oxide thin films. Typically, such oscillations are attributed to the layer-by-layer growth of films, but so far there were some knowledge gaps referring to some unusual features observed in the patterns, e.g., unexpected frequency doublings or sign reversals and phase shifts in the RHEED oscillations.

This excellent work brings new knowledge in the origin of such unusual features in the RHEED oscillations and provides invaluable and clear guidelines about using RHEED to achieve atomic control of oxide films. This is an all-important aspect, as it is known that many emergent functionalities are very critical on the atomic-scale control of oxide heterointerface. A clear evident example of this is presented by the authors in this manuscript, by applying their guidelines to the growth of LAO/STO interfaces.

To my view, this manuscript is very well written, with an interest for a vast community working in oxides and with plenty of priceless insights into the topic of thin film growth. I am convinced that it deserves publication.

I have just a couple of minor remarks or comments:

a) The model is presented exhaustively in the Supplementary File. Yet, I couldn't see what values did the authors choose for the f_j factors (atomic scattering factors). Did the authors use tabulated values or were they instead calculated.

b) Concerning the modelling of the RHEED patterns, I understand that the basic approach using kinematic theory can explain the basic features observed in the RHEED experiments. Nevertheless, I would appreciate a deeper discussion about dynamical theory which is probably better suited to understand RHEED.

Reviewer #2:

Remarks to the Author:

Using experiments with RHEED oscillation and model calculations, the authors present an interesting way to explain and monitor the layer-by-layer growth of SrTiO₃ (STO) thin films which relies on the variation of the mean inner potential by analyzing the Kikuchi lines during growth. The paper indeed brings additional information to the community working on these type of interfaces and growth techniques, however, these findings are still not considered by me as a key contribution to the current understanding of how to grow high quality films. Even though the experimental measurements with RHEED seem to be of a high quality, the interpretation of this manuscript is based only on the experiments using RHEED in my opinion, this required to be supported/verified also by other techniques, e.g. HRTEM, XPS, AFM and transport. The paper also lack a description/comparison to other available in-situ techniques to monitor the chemistry of the growing surface such as, in-situ XPS, time-dependent lattice parameter using RHEED, in-situ plasma diagnostics, in situ electron diffraction etc. Overall, I am not convinced that this paper considerably advances our knowledge in growing oxides interfaces and the manuscript does therefore not meet the criteria for publication in Nature Communication. Therefore, I feel that this paper would find a more appropriate outlet in more specialised journal.

Besides the above, there are still a few open questions that need to be answered to improve the manuscript:

1. The author claim in their motivation for the work that " As a consequence of the poor understanding of the origin of RHEED oscillations and the lack of in situ measurements of the chemical information..", RHEED oscillation has been investigated for many years and reported in numerous of manuscripts and books etc. So, saying that the origin of the RHEED oscillation is poorly understood is not completely correct. Beside, in-situ chemical information during growth have also been reported using other techniques, this has not been commented in the manuscript.
2. The authors claim that "With a large amplitude, the frequency doubling feature due to the surface roughing is minimised..." This statement was never confirmed experimentally in this manuscript using another independent technique e.g. AFM or HRTEM. I found that this and many statements would need a much more careful analysis to be made convincing by using other techniques.
3. The authors said that "The capability of synthesizing 2DEL on arbitrary thick STO epitaxial films underscores the validity....." As the authors surly know, high-quality SrTiO₃ films where already reported previously e.g. on STO (Journal of Vacuum Science & Technology A, 27, 461 (2009)) or on Si (APL MATERIALS 2, 116109 (2014) and Nature Communications 1, Article number: 94 (2010)). How can one judge the quality of the current films? What was the mobility of the 2DEL and how this compared to previous reported values? This could help evaluating the quality of the films.

Reviewer #3:

Remarks to the Author:

Mao and colleagues try to resolve a problem that troubles many thin film growers: how to interpret the RHEED oscillations and can one determines the exact termination using these oscillations? From a fit to the Kikuchi lines they extract the inner potential, and by incorporating it into the calculation they were able to capture the dependence of the phase of the oscillations on incident angle. The authors relate the doubled frequency component to inelastic scattering off incomplete surface while the main contribution to the observed oscillations comes from the oscillating surface potential.

To demonstrate their control the authors grow various layers of LaAlO₃ on both SrO terminated SrTiO₃ and TiO₂ terminated one as designed by their interpretation of the RHEED oscillations during SrTiO₃ deposition. The results are consistent with previous observations of critical LaAlO₃ thickness for conductivity and no conductivity for the SrO terminated SrTiO₃ interface. The authors summarize with three thumb rules that help a broad audience of film growers to optimize their films according to their RHEED pattern and its time evolution.

The paper is very well written with enlightening figures and explanations. Before publications I would like the authors to address the following questions:

1. The analysis is incompatible with films that are different than the substrate. For instance LaAlO₃ on SrTiO₃. This is because the mean free path of the electrons is 11nm (or ~6Å along the z direction) and hence the inner potential changes during the deposition of the first 2-4 LaO/AlO₂ monolayers. Can the authors comment on this point?
 2. The error bars in figure 2C are rather large and it seems that a horizontal line can go through the data as well. How reliable is the fit and what is the reason for the large error bars?
- Other than that I would recommend publication.

Response to Referee #1:

We thank Referee #1 for the thorough review and the recommendation of publication. Below we address Referee A's questions and concerns point by point.

Q1: The model is presented exhaustively in the Supplementary File. Yet, I couldn't see what values did the authors choose for the f_j factors (atomic scattering factors). Did the authors use tabulated values or were they instead calculated.

As it is widely employed in the literature (Ref [42]), the atomic number Z is also used as the f_j factors (atomic scattering factors) in our manuscript considering the scattering vector equals zero. It was written in the supplemental material where it may be easy to be missed. In order to present this information more clearly, we have revised the sentence in the supplemental section and also added the explanation of f_j factors in the main text.

In Supplementary section, Page 7 (text in blue): "*For atomic scattering factor, when scattering vector $s=0$, f_j equals to the atomic number Z (Sr is 38, Ti is 22 and O is 8) [13].*"

In main text, Page 4 (text in blue): "*where, f_j is the atomic scattering factor [42], for Sr is 38, Ti is 22 and O is 8, ...*"

Ref [42]: A. Ichimiya and P. I. Cohen, *Reflection high-energy electron diffraction* (Cambridge University Press, 2004).

Q2: Concerning the modelling of the RHEED patterns, I understand that the basic approach using kinematic theory can explain the basic features observed in the RHEED experiments. Nevertheless, I would appreciate a deeper discussion about dynamical theory which is probably better suited to understand RHEED.

We thank Referee #1 for the comment that the kinematic theory explains the features observed in the RHEED experiments. As suggested by the referee, we try to give a deeper discuss of the dynamical theory. The dynamical theory treats the diffraction as a multiple scattering process and considers the oscillation mainly as a consequence of contributions of the diffuse scattered potential components. The kinematic theory only considers the single scattering process and the mean inner potential relates to the sum of the atomic potential. The dynamical theory is quantitatively more accurate if the approximations describe the real system properly. Nonetheless, the multi-slice method and the complex algebra involving the recursion of transfer scattering matrix make the simulation using dynamical theory a rather difficult task and it is also difficult to visualize the key underlying physics, such as the role of mean inner potential. Considering we can directly measure the mean inner potential during the growth, kinematic theory is more suitable to

investigate the role of the mean inner potential in determining RHEED intensity.

We have revised the sentence in the main text:

In main text, Page 4 (text in blue):

Original version: *“Electron diffraction is fundamentally a dynamical effect involving multiple scattering processes, making it an extremely difficult task to simulate the exact diffraction intensity. In the following discussion, ...”*

Revised version: *“Electron diffraction is fundamentally a dynamical effect involving multiple scattering processes. The complex algebra involving the recursion of transfer scattering matrix makes the simulation using dynamical theory a rather difficult task and it is also difficult to visualize the key underlying physics, such as the role of mean inner potential. Considering we can directly measure the mean inner potential, kinematic theory is the more suitable model to investigate the role of mean inner potential in determining the RHEED intensity. In the following discussion, ...”*

Response to Referee #2:

We thank Referee #2 for considering our work to be “interesting” and “brings additional information to the community”. We also thank Referee #2 for his/her careful reading of our paper and the excellent and constructive feedbacks, which help us improve the manuscript significantly. We do realize that our previous version did not explain our model and contribution very clearly. Following Referee #2’s suggestions, we have revised the manuscript substantially by including a lot more new data and discussions, which explains the contribution of our work more clearly.

Below we summarize Referee #2’s questions and concerns and address them point-by-point.

Q1: *“The paper indeed brings additional information to the community working on these type of interfaces and growth techniques, however, these findings are still not considered by me as a key contribution to the current understanding of how to grow high quality films...Overall, I am not convinced that this paper considerably advances our knowledge in growing oxides interfaces.”*

We thank the referee for considering our work “brings additional information to the community”. The application of *in situ* RHEED has fundamentally advanced the epitaxial thin film growth with atomic precision. RHEED intensity oscillations have been explored and well understood in the growth of conventional semiconductors. In contrast, RHEED oscillations of complex oxides are very complicated and sometimes controversial as reported in the literature. For example, the number of RHEED oscillations typically corresponds to the number of grown repeated units, but

it is unclear whether the maximum or minimum intensity is corresponding to the full surface termination in the growth of complex oxides. As oxide interfaces host a wide variety of novel and exciting quantum phases, deeper understanding of RHEED oscillations and general criteria to guide the synthesis of chemically sharp oxide surfaces and interfaces is of great importance.

As both Referee #1 and Referee #3 also consider our work to be important for the community, such as “*provides invaluable and clear guidelines*” and “*help a broad audience*”, we strongly believe that our work indeed advances the understanding of the complicated RHEED oscillations in the growth of complex oxides, which can help the chemically precise growth of oxide interfaces. More specifically, our work has the following key contributions:

1. Our work reveals the strong incident angle dependence of the RHEED oscillations in the growth of SrTiO₃, a prototype perovskite oxide. This observation reconciles the contradicting observations reported in the literature and unambiguously shows that incident angle should be considered and properly optimized in applying RHEED oscillations to control the oxide surface termination. Terminating the growth at minimum intensity or maximum intensity with random incident angle can not guarantee the synthesis of chemically sharp surface/interface.
2. By fitting Kikuchi lines to extract the mean inner potential directly, we found that the mean inner potential of SrTiO₃ shows strong surface dependence, which provides *in situ* and *real-time* chemical information of the growth front of complex oxides. With this information, one can monitor and control the surface termination of oxide films easily. This real-time measurement has big advantages over other *in situ* but not real-time techniques.
3. Based on experimental observations and theoretical simulations, we propose a model to successfully explain the origin of the complicated RHEED oscillations in the growth of complex oxides, where mean inner potential, surface roughness and incident angle of the electron beam should be considered simultaneously.
4. We propose three general guidelines to help the precise control of oxide surface/ interface, which can be applied in MBE, PLD and other techniques.
5. Last but not the least, following our guidelines, we demonstrate that the TiO₂-termination of arbitrary thick as-grown SrTiO₃ films can be of as high quality as pre-etched SrTiO₃ substrates. Following Referee #2's suggestions, XRD, AFM, LFM, STEM, XPS measurements have been performed and the data are now added in the revised manuscript. Note that all STO films in our work were grown under normal layer-by-layer growth condition (typical temperature, press, etc.) and there is no need for post annealing, high growth temperature, volatile species or metal-organic precursor, etc.

One last thing we want to clarify is that we did not try to claim that our method is the only way to synthesize high quality oxide interfaces and would like to apologize if the previous version of the manuscript somehow delivers this wrong message. We are sure aware of the high quality oxide

interfaces using optimized growth receipts by other groups using many techniques, including, PLD, MBE and hybrid MBE. What we try to present here is the deeper understanding of complicated RHEED oscillations in the growth of complex oxides, direct measurements of the mean inner potential and its important application in determining the surface termination of oxide films. With these knowledge and general guidelines, one can easily control the surface termination and oxide interfaces with the proper application of RHEED. To clarify this, we have added the following discussion in the main text, Page 4 (text in blue just before the conclusion section):

“Note that high quality oxide surface/interface and high mobility 2DEG can also be synthesized with special growth receipts including post annealing, high growth temperature, or metal-organic precursor, etc [32-34, 39]. Our guidelines provide a way to control the surface termination and oxide interface under normal layer-by-layer growth condition with the proper application of RHEED.”

In the revised manuscript, we have also included a lot more new data and discussions and more details can be seen in the answer to the following specific questions.

Q2: *“Even though the experimental measurements with RHEED seem to be of a high quality, the interpretation of this manuscript is based only on the experiments using RHEED in my opinion, this required to be supported/verified also by other techniques, e.g. HRTEM, XPS, AFM and transport.”*

Following Referee #2’s suggestion, we have performed AFM, LFM (lateral force microscopy), XRD, XPS, STEM and transport measurements on our samples and the data are shown in the revised supplementary information Fig. S1-Fig. S3 (also shown below). The AFM measurement shows clear terraces and the LFM measurement show no clear contrast, indicating our as-grown films have the same high quality pure TiO₂ termination as that of etched STO substrates. The XRD data show no thickness fringes and the rocking curve is as sharp as the pure STO substrate, indicating the grown films cannot be distinguished from the STO substrate. In the XPS data, the Sr/Ti intensity ratio of SrO-terminated STO film and TiO₂-terminated STO film are consistent with expectation. The STEM data of the LAO/STO interface grown on STO film show atomically abrupt LaO/TiO₂ interface, which is comparable with the LAO/STO interface grown directly on etched STO substrates by other groups. The transport measurements of the 2DEG of our LAO/STO (film) show a mobility of 4,591 cm²/Vs at 5K, which is similar to those reported. Also note that the 2DEG strongly depends on the LAO stoichiometry (perfect LAO films grown on STO shows no 2DEG (Warusawithana et al., *Nature Communications* **4**, 2351 (2013))). We believe higher mobility 2DEG can be achieved if we have more time to optimize the LAO stoichiometry but it is not the main purpose of this work.

All of these measurements are consistent and of high quality. We have revised our manuscript accordingly:

In the main text, Page 2 (text in blue): “*The quality of film surfaces and interfaces was also characterized by other techniques, including high-resolution transmission electron microscopy (HRTEM), X-ray diffraction (XRD), X-ray photoelectron spectroscopy (XPS), atomic force microscopy (AFM) and lateral force microscopy (LFM). All of the data are of high quality and consistent with our conclusions.*”

In addition, we have substantially revised the supplementary information.

In supplementary information, page 1-5, we have added the data and analysis of all characterization measurements.

Supplementary Figure S 1: **Ex situ characterizations of as-grown STO films.** (a), High-resolution θ - 2θ XRD scan of 20 u.c. STO film with TiO_2 -termination. (b), The fine scans around (002) STO peak of the stoichiometric and off-stoichiometric STO films. (c), AFM, (d), LFM and (e), XPS measurements of the same TiO_2 -terminated STO film.

Supplementary Figure S 2: **STEM-ADF images and elemental sensitive intensity analysis of LAO/STO interface.**

Supplementary Figure S 3: **Transport measurements of the 2DEG grown on SrTiO₃ films.**

Q3: *The paper also lack a description/comparison to other available in-situ techniques to monitor the chemistry of the growing surface such as, in-situ XPS, time-dependent lattice parameter using RHEED, in-situ plasma diagnostics, in situ electron diffraction etc.*

We thank the referee for this suggestion. Following the suggestion, we analyzed the *in situ* time-dependent lattice parameter using RHEED taking during the layer-by-layer growth of SrTiO₃ films. We found that there is no clear time dependence of the lattice constants since there is no lattice relaxation during the homoepitaxial film growth of arbitrary thickness. For XPS measurements, we don't have the capability of *in situ* XPS measurement in our lab. Instead, we have performed *ex situ* XPS measurements on TiO₂-terminated and SrO-terminated SrTiO₃ films as shown in above Fig. S1. The *ex situ* XPS results support our conclusion of the surface termination control. This result is now added in the revised version of the supplementary material. We also want to point out that *in situ* XPS measurements we know of could not work during the growth at high growth pressure in real time. In contrast, monitoring the surface termination in real time is a big advantage of RHEED technique. Also, we try our best to search for but could not find too much related information of *in-situ* plasma diagnostics mentioned by the referee, so we could not comment on that. Regarding the *in situ* electron diffraction, we are not quite sure which type of electron diffraction the referee referred to. RHEED is actually an electron diffraction technique, which is under studied in this work. Another common electron diffraction technique is low energy electron diffraction (LEED). LEED also cannot work at the growth pressure and it is also difficult to tell the chemical information of the surface layer.

As mentioned in the answer to Q2, we have added XPS measurement results in the supplemental information (Supplementary Fig. S1 (f)).

Q4: *The author claim in their motivation for the work that “ As a consequence of the poor*

understanding of the origin of RHEED oscillations and the lack of in situ measurements of the chemical information..”, RHEED oscillation has been investigated for many years and reported in numerous of manuscripts and books etc. So, saying that the origin of the RHEED oscillation is poorly understood is not completely correct. Beside, in-situ chemical information during growth have also been reported using other techniques, this has not been commented in the manuscript.

We agree with the referee that this statement can cause some confusion and we should be more specific in describing the existing problem. We fully agree that RHEED oscillations for many material systems are well understood, especially in the growth of semiconductor films. What we really meant here is that the complicated RHEED oscillation patterns observed in the literature during the layer-by-layer growth of complex oxide films are complicated and sometimes controversial as reported in the literature. For example, although the thickness control of the film growth can be in atomic level, however, it is unclear if one should stop the growth at maximum or minimum intensity to obtain pure termination, i.e. TiO₂-termination in the case of SrTiO₃.

Upon the referee’s suggestion, we have revised the statement (Page 1, text in blue):

Original version: *“As a consequence of the poor understanding of the origin of RHEED oscillations and the lack of in situ measurements of the chemical information, it is challenging to obtain chemically sharp oxide interface.”*

Revised Version: *“Since the RHEED oscillations during the layer-by-layer growth of complex oxide films are complicated and sometimes controversial as reported in the literature [19, 22-25, 29], commonly accepted criteria for real-time control of the surface termination by RHEED is still lacking and it is very challenging to synthesize chemically sharp oxide interface.”*

We also agree with the referee that chemical information can also be measured by other *in-situ* techniques, such as X-ray photoemission spectroscopy (XPS). Nevertheless, XPS typically needs UHV and not applicable to provide *real time* information. In contrast, monitoring the surface termination in real time is a big advantage of RHEED technique. Following Referee #2’s suggestions, we have performed many *ex situ* measurements to confirm the high quality control of our film surfaces/interfaces as shown in the supplementary information.

In the revised manuscript, we also add this corresponding comment.

In main text, Page 2 (text in blue): *“Although other in situ measurements, such as X-ray photoemission spectroscopy (XPS), can provide chemical information of the surface termination [30,31], most of these techniques need ultrahigh vacuum and not applicable in monitoring the chemical composition in real time during the growth.”*

Q5: *The authors claim that “With a large amplitude, the frequency doubling feature due to the surface roughing is minimized...” This statement was never confirmed experimentally in this manuscript using another independent technique e.g. AFM or HRTEM. I found that this and many statements would need a much more careful analysis to be made convincing by using other techniques.*

We thank the referee for pointing out this confusing expression. In fact, we are not comparing different film growths and claim that the growth with larger RHEED oscillation amplitude can yield better termination. What we want to say here is that it is easier to determine when to stop the growth in order to obtain pure termination if the RHEED oscillation amplitude is large by optimizing the electron beam incident angle. This is just a straight interpretation of the experimental data shown in Fig. 1e and the simulated data in Fig.3e. In Fig. 1e, the oscillation curve taken at 1.92° has larger amplitude due to the larger diffraction intensity difference from SrO and TiO₂ terminations. At 2.28° incident angle, the diffraction intensity difference from SrO and TiO₂ terminations are zero and the frequency-doubled oscillation is mainly due to the roughness variation of the surface layer. With the contribution of frequency-doubled oscillation pattern, there are two minima in the growth of one unit cell and none of them corresponds to pure termination. For large oscillation amplitude, the minima correspond to the pure termination and thus easier to determine when to stop the growth.

To avoid the confusion, we have revised the expression:

In main text, page 6 (text in blue):

Original version: *“Rule 2: The larger the amplitude of the RHEED oscillations, the more precise the control of the termination. With a large amplitude, the frequency doubling feature due to the surface roughing is minimized and the RHEED maxima or minima are more likely to correspond to pure surface terminations.”*

Revised Version: *“Rule 2: The larger the amplitude of the RHEED oscillations by optimizing the incident angle of the electron beam, the easier to obtain precise control of the termination. As shown in the experimental data and in the model simulation, the oscillation amplitude strongly relies on the incident angle of the electron beam. At certain incident angles, two different terminations yield similar diffraction intensity where the total oscillation amplitude is small and the frequency is doubled. Thus, the minima do not correspond to any full terminations. By optimizing the incident angle of the electron beam to maximize the total oscillation amplitude, both maxima and minima correspond to pure terminations, and it is easier to obtain more precise termination control.”*

Q6: *The authors said that “The capability of synthesizing 2DEL on arbitrary thick STO epitaxial films underscores the validity.....” As the authors surely know, high-quality SrTiO₃ films were already reported previously e.g. on STO (Journal of Vacuum Science & Technology A, 27, 461*

(2009)) or on Si (*APL MATERIALS* 2, 116109 (2014) and *Nature Communications* 1, Article number: 94 (2010)). How can one judge the quality of the current films? What was the mobility of the 2DEL and how this compared to previous reported values? This could help evaluating the quality of the films.

We thank the referee for these useful suggestions. We have cited these articles in the revised version of the manuscript. In our manuscript, we are not claiming that our film quality is much better than those grown by other groups using special receipt, such as post annealing, high growth temperature or volatile species, etc. Here, we mainly want to report the deeper understanding of the RHEED and the real time measurement of the mean inner potential it application of the chemically precise control of the oxide surface/interface with normal growth condition. Realizing 2DEG by growing LAO on as-grown STO film is just one of the many proofs to demonstrate that the as-grown STO films are indeed terminated by TiO₂ layer.

In the *Nature Communications* article mentioned by the referee, the metallic LAO/STO interface fabricated on Si substrate needs the thermal annealing of the STO film after growth to improve the structural quality. In contrast, our method can yield pure termination of the as-grown film under normal growth condition.

In the *Journal of Vacuum Science & Technology* article mentioned by the referee, high quality SrTiO₃ films were grown by a hybrid molecular beam epitaxy approach. There is no discussion of the termination control in the article. Since the metal-organic precursor is used and the growth is at thermodynamic regime, the film may have preferential surface termination.

In the *APL MATERIALS* article mentioned by the referee, they studied the LaTiO₃/SrTiO₃ interface. Since the B-sites are the same, either SrO or TiO₂ terminated STO should give similar interface structure. It is not that critical to control the TiO₂ termination in that case. Also, the STEM image is sort of blurry and it is difficult to judge how sharp the interface is.

In the revised manuscript, page 2 (text in blue), we have cited the above articles in the main text as [32], [33] and [39]:

“To date, 2DELS in LAO/STO have mainly been fabricated on chemically etched TiO₂-terminated SrTiO₃ substrate, post-annealed SrTiO₃ films or silicon substrates [1, 2, 32, 33].”

“Ex situ etching and annealing after the growth, growing SrTiO₃ films in very high temperature (1100°C), and using metal-organic precursors in hybrid MBE have been shown to achieve TiO₂-terminated surface [36–39] ...”

Following the referee’s suggestions, we also characterized our film quality by AFM, XRD, XPS, STEM-ADF and transport measurements, and added the new data in the supplementary

information. All data are consistent and of high quality, indicating the chemically specific control of the surface termination and interface configurations.

The transport measurements of the 2DEG of our LAO/STO (film) show a mobility of 4,591 cm^2/Vs at 5K, which is similar to those reported. Also note that the 2DEG strongly depends on the LAO stoichiometry (perfect LAO films grown on STO shows no 2DEG (Warusawithana et al., *Nature Communications* **4**, 2351 (2013.)). We believe higher mobility 2DEG can be achieved if we have time to optimize the LAO stoichiometry but it is not the main purpose of this work. (The answer is the same as **Q2**)

More details can be seen in supplementary information Fig. S1-Fig. S3.

Response to Referee #3:

We thank Referee #3 for the thorough review and the favorable comments. We appreciate Referee #3 for recognizing the importance of our work for the oxide film growth.

Below we respond to Referee #3's comments and concerns:

Q1: The analysis is incompatible with films that are different than the substrate. For instance LaAlO_3 on SrTiO_3 . This is because the mean free path of the electrons is 11nm (or $\sim 6\text{\AA}$ along the z direction) and hence the inner potential changes during the deposition of the first 2-4 LaO/AlO_2 monolayers. Can the authors comment on this point?

We thank Referee #3 for pointing out this very important point. We agree with the referee that the different diffraction structure factors and mean inner potential between substrate and film can yield complicated RHEED oscillation pattern at the initial growth. This actually explains why the RHEED oscillation patterns are typically complicated at the initial growth of heteroepitaxial film. In the revised version of the manuscript, we have added the corresponding discussion.

In the main text, Page 6 (text in blue): “*For heteroepitaxial film growth, the initial oscillation patterns are typically complicated due to the different diffraction structure factor and mean inner potential. Only when the film is thicker than the mean free path of the electron beam, stable RHEED oscillation pattern can be achieved and the above method can be applied.*”

Q2: The error bars in figure 2C are rather large and it seems that a horizontal line can go through the data as well. How reliable is the fit and what is the reason for the large error bars?

We agree with the referee that the error bar in figure 2c (in the main text) is rather large. In fact,

the big error bar is only for the systematic error of the *absolute* value of the mean inner potential, but the error bar for the more meaningful *relative* mean inner potential is actually fairly small. In the real experimental setup, the diffraction geometry may not be perfect. For example, RHEED screen is not perfectly perpendicular to the electron beam. Also, the RHEED images taken by CCD may be slightly distorted. As such, the line shape of the Kikuchi lines is not perfectly accounted by the ideal equation, which gives relatively large systematic error in the data fitting. However, the Kikuchi line shifts its position during the film growth, which corresponds to the relative change of the mean inner potential during the growth. The fitting error bar of this position shift is much smaller, indicating the relative change of the mean inner potential is real and reliable.

In Fig.2 (main text, Page 3), we have inserted in panel c the clear position shift of the Kikuchi lines, showing the small fitting error bar and reliable change of the relative mean inner potential.

FIG. 2: **Kikuchi lines and atomic layer-by-layer variations of the mean inner potential.** (a), Schematic of the formation of Kikuchi lines. G is the reciprocal vector of the lattice, P and D are the momentum vectors of the primary incident electron beam and the outgoing beam that satisfy the Bragg condition. The red dots and lines on the screen are the diffraction peaks and the Kikuchi lines, respectively. (b), A typical RHEED pattern taken along the SrTiO₃ [110] direction, showing Kikuchi lines clearly. The dashed lines are fits to the Kikuchi lines using Eq. (1). (c), The upper panel shows the periodic variation of the extracted mean inner potential. The lower panel shows the position shift R_y of the Kikuchi lines and the small error bar indicates clear and reliable relative change of the mean inner potential during the growth.

In main text, Page 4 (text in blue), we explain the large (small) error of the absolute (relative) mean inner potential.

“Note that the error bar of the absolute mean inner potential is relative large. This large systematic error is because the lineshape of the Kikuchi lines could not be perfectly accounted by ideal equation due to the imperfect diffraction geometry and distortion of the RHEED images taken by the CCD camera. Nonetheless, the position shift of the Kikuchi lines also oscillate during the growth and the small fitting error bar indicates the relative change of the mean inner potential is clear and reliable (Fig. 2c).”

In addition, we also insert a new section VII and Fig. S7 in the Supplementary information (page 11-12) to explain this issue in detailed.

Reviewers' Comments:

Reviewer #1:

Remarks to the Author:

To my view, in the new correspondence the authors have replied satisfactorily the minor remarks that I raised previously. After reading the revised manuscript and the changes therein, I am convinced that the work deserves publication. Overall, it represents an important advance in the knowledge about the atomic control of oxide film growth, which is extremely relevant for the achievement of novel functionalities emerging at oxide heterointerface. It provides important guidelines and should pique the interest of a large community working in oxides, with important insights into the topic of thin film growth.

Reviewer #2:

Remarks to the Author:

The authors have performed careful additional measurements which comply with most of my comments/suggestion. Again, the HREED measurements appear to have in situ observation of atomic layer-by-layer during the epitaxial growth yielding information on the chemical composition of the surface layer during thin films growth. This point alone is truly new and novel, suggesting that the paper should be published somewhere but, in my opinion, the current results do not advance the field to the extent that it would warrant publication in a high impact journal were a broad audience will be interested at.

Reviewer #3:

Remarks to the Author:

I read the revised manuscript and the rebuttal letter. The authors have properly addressed all the Referee comments and criticisms and the paper has significantly improved. I disagree with Referee B, who also gave very useful remarks, and I think that the paper should be published in Nature Communications in its present form.